# Formation and Inhibition Mechanism of Na_8_SnSi_6_O_18_ during the Soda Roasting Process for Preparing Na_2_SnO_3_

**DOI:** 10.3390/ma15248718

**Published:** 2022-12-07

**Authors:** Zijian Su, Shuo Liu, Benlai Han, Yuanbo Zhang, Tao Jiang

**Affiliations:** School of Minerals Processing and Bioengineering, Central South University, Changsha 410083, China

**Keywords:** Na_2_SnO_3_, Na_8_SnSi_6_O_18_, phase transformation, TG-DSC analysis

## Abstract

To produce Na_2_SnO_3,_ which is widely used in the ceramics and electroplating industries, a novel process for the preparation of sodium stannate from cassiterite concentrates was developed successfully by the authors’ group. It was found that sodium stannosilicate (Na_8_SnSi_6_O_18_) was easily formed due to the main gangue of quartz in cassiterite concentrates, which was almost insoluble and decreased the quality of Na_2_SnO_3_. The formation and transitions of Na_8_SnSi_6_O_18_ in the SnO_2_–SiO_2_–Na_2_CO_3_ system roasted under a CO–CO_2_ atmosphere were determined. The results indicated that the formation of Na_8_SnSi_6_O_18_ could be divided into two steps: SnO_2_ reacted with Na_2_CO_3_ to form Na_2_SnO_3_, and then Na_2_SnO_3_ was rapidly combined with SiO_2_ and Na_2_CO_3_ to form low melting point Na_8_SnSi_6_O_18_. In addition, Na_8_SnSi_6_O_18_ can be decomposed into Na_2_SiO_3_ and Na_2_SnO_3_ by using excess Na_2_CO_3_.

## 1. Introduction

Na_2_SnO_3_ is an important raw material to produce stannate ceramics and electroplating materials [1,2,3,4,5]. A novel soda roasting–leaching process has been developed by the authors’ group using cassiterite concentrates as raw materials, by which Na_2_SnO_3_ was prepared efficiently and cleanly [6]. Previous studies have showed that the solid-state reactions between SnO_2_ and Na_2_CO_3_ are accelerated under a CO–CO_2_ atmosphere [7,8,9,10]. Then, a trihydrate sodium stannate product with high purity was obtained, which meets the requirements of an industrial first-grade product [6]. In addition, the soda roasting process has also been applied for the comprehensive utilization of tin-bearing secondary waste [11,12,13,14].

Cassiterite (SnO_2_) is the primary source of tin. It is naturally formed by magmatic-hydrothermal processes and occurs in granite pegmatites, quartz veins, greisens associated with granites, highly fractionated granites, as well as placer deposits [15,16,17,18]. Nevertheless, gangue minerals, including calcite, magnetite and other oxides, cannot be perfectly separated by beneficiation combined methods. Hence, oxidizing roasting and hydrochloric acid leaching processes are applied to remove impurity elements (Fe, Ca, Mg, S, As, Pb, Zn, etc.). However, quartz is stubborn and difficult to remove during the pretreatment process, resulting in the residue of SiO_2_ in tin concentrates being as high as 8 wt.% [19,20].

Cassiterite concentrate, as reported, seldom reacts with soda (Na_2_CO_3_) under air atmospheres. However, the authors’ group found in previous research that cassiterite (or SnO_2_) could readily react with Na_2_CO_3_ under an appropriate CO–CO_2_ atmosphere. In the roasting process, the CO gas molecules were firstly adsorbed on the SnO_2_ surface and then combined with the bridging oxygen so that some oxygen vacancies were formed. These vacancies were replenished by the active oxygen anions in Na_2_O, which was the decomposition product of Na_2_CO_3_ roasted over 851 °C. These processes accelerated the formation of Na_2_SnO_3_ [6,7,8,9,10].

Our previous studies have found that the quartz (SiO_2_) in the raw material has a significant effect on the phase transformation of SnO_2_ during the soda roasting process [21]. The target products of soda roasting were Na_2_SnO_3_ and Na_2_SiO_3_, which are freely soluble in NaOH solution. It was found that Na_8_SnSi_6_O_18_ was easily formed and was almost insoluble during the leaching process, which decreased the recovery of tin [6,21]. A series of studies have systematically revealed the reaction mechanism of SnO_2_–SiO_2_, which was investigated during the cassiterite reduction smelting process and flat glassmaking method [22,23,24]. Furthermore, those studies confirmed that SnO_2_ was an acidic oxide, while it transformed into SnO, an alkali oxide, during the reduction process. However, no studies have mentioned the reactions in the SnO_2_–SiO_2_–Na_2_CO_3_ system, especially under a CO–CO_2_ atmosphere.

Based on our previous studies, SiO_2_ in cassiterite concentrates has adverse effects on the formation and leaching of Sn during the soda roasting–leaching process. The maximum conversion rate of Sn was around 85.6% under optimal conditions. However, the formation and phase transformation mechanisms of Na_8_SnSi_6_O_18_ were unknown. Hence, in order to improve the Sn conversion rate during the soda roasting process, the formation mechanism and decomposition process of Na_8_SnSi_6_O_18_ in the SnO_2_–SiO_2_–Na_2_CO_3_ system were investigated, using X-ray powder diffraction (XRD), scanning electron microscopy and energy dispersion spectroscopy (SEM–EDS), thermogravimetric and differential scanning calorimetry (TG-DSC), Fourier transform infrared spectroscopy (FTIR), etc. 

## 2. Experimental

### 2.1. Materials

The cassiterite concentrates (taken from Gejiu, Yunnan Province of China, Yunnan Tin Company Limited) used in this study were pretreated by oxidizing roasting and acid leaching processes to remove impurities [6,13]. As shown in Figure 1, only diffraction peaks of cassiterite (SnO_2_) and quartz (SiO_2_) were found in the XRD pattern of the pretreated cassiterite concentrates. In addition, the contents of Sn and Si were determined to be 62.93 wt.% and 3.66 wt.% by ICP-AES, respectively, while impurities of Ca, Fe, Mg, Al and S were not detected. Moreover, the analytical reagents of SnO_2_, SiO_2_, Na_2_CO_3_, Na_2_SiO_3_ and Na_2_SnO_3_·3H_2_O (AR, Shanghai Aladdin Bio-Chem Technology Co., Ltd., Shanghai, China) used in this study had a purity of over 99.5 wt.%, and all the samples were pre-ground to fully pass through a 0.037 mm sieve. The gases CO, CO_2_ and N_2_ had a purity of 99.9 vol%.

### 2.2. Methods

#### 2.2.1. Experimental Procedures

The experimental procedures in this study mainly include the roasting and leaching process, where the details of the roasting process under a 15 vol% CO/(CO and CO_2_) atmosphere have been described in our previous study [6,21]. The leaching of Na_2_SnO_3_ tests were conducted in a water bath at 40 °C with a content of 0.05 mol/L NaOH solution. Finally, the leaching solution was filtered and prepared to determine the formation efficiency of Na_2_SnO_3_. The residues were washed with distilled water to identify the phase constituents as follows:(1)$$L=\frac{1000\mathrm{CV}}{MW}\times 100\%$$
where *L* is the formation efficiency of Na_2_SnO_3_, *M* is the weight of the roasted samples (g), *W* is the grade of Sn in the roasted samples (%), C is the mass content of Sn in the leaching solution (mg/mL) and V is the volume of leaching solution (mL).

#### 2.2.2. Instrument Techniques

The phase constituents of the samples were identified by X-ray diffraction XRD (Cu-target Bruker D8 Advance), with a step of 0.02° at 10 min^−1^ ranging from 10° to 80°. The microscopic morphology was observed with a scanning electron microscope (QUANTA 200, FEI, Eindhoven, The Netherlands) equipped with an EDAX energy dispersive X-ray spectroscopy (EDS) detector (EDAX Inc., Mahwah, NJ, USA). Fourier transform infrared spectroscopy (FTIR: Nicolet 8700) in the range of 400–4000 cm^−1^ was applied to determine the chemical bands of the roasted samples in transmission mode. TG-DSC analyses of samples were performed using a thermal analyzer (Netzsch STA 449, Selb, Germany) in the temperature range of 25–1200 °C with a heating rate of 10 °C/min in an Ar atmosphere, and a platinum crucible was used with 50 mg samples for each test. The content of Sn in the solid material and the aqueous solution was determined using inductively coupled plasma atomic emission spectroscopy (ICP-AES; Thermo Fisher Scientific, Waltham, MA, USA, Icap7400 Radial, King of Prussia, PA, USA). For each ICP test, a certain mass or volume of samples was first dissolved in H_2_SO_4_–HF solution system, and the solution was set to a constant volume of 100 mL. Then, the solution was tested using ICP, and the Sn content in the raw solid material and the aqueous solution were calculated. The morphological evolution of the roasted samples was monitored by an in situ high temperature thermal analyzer (S/DHTT-TA-III, Chongqing University, Chongqing, China).

## 3. Results and Discussion

### 3.1. Phase Analysis for the Products of Soda Roasting

Figure 2 shows the experimental flowsheet of soda roasting and leaching using Si-bearing cassiterite concentrates. Based on a previous paper, the optimal experimental conditions were fixed at a roasting temperature of 875 °C, a CO content of 15%, roasting time of 15 min, Na_2_CO_3_/SnO_2_ mole ratio of 1.5, etc. [6,21]. A Sn leaching efficiency of 85.6% was achieved; moreover, a small number of leaching residues were obtained. Then, the roasted products and leaching residues (in Figure 2) were observed by XRD and SEM-EDS analysis, and the results are shown in Figure 3.

As shown in Figure 3a, the main phases in the roasted products were Na_2_SnO_3_ and Na_2_SiO_3_, which verified the high leaching efficiency of Sn and Si. In particular, the characteristic peaks of Na_8_SnSi_6_O_18_ were found in the leaching residues of Figure 3b, and unreacted SnO_2_ was observed as well. It is seen from Figure 3c that Na_2_SnO_3_ (Spot A in Figure 3c) was found as regular hexagonal cylinder crystal grains and slice crystal grains, which matched well with the theoretical molar ratio of sodium stannate of 2:1. Moreover, melting phases can be seen in the backscattering image of Figure 3c, which were formed irregularly at the grain edge of Na_2_SnO_3_. The EDS analysis of Spot B in Figure 3c showed that the main element composition was Sn, Na and Si, which is consistent with the chemical composition of Na_8_SnSi_6_O_18_. The morphology of leaching residues is shown in Figure 3d. The results indicated that the Na_8_SnSi_6_O_18_ phase (Spot C in Figure 3d) was closely wrapped in cassiterite particles. Both SnO_2_ and Na_8_SnSi_6_O_18_ were insoluble during the leaching process, and then they were enriched in the leaching residues. The results in Figure 3 indicate that Na_8_SnSi_6_O_18_ was rapidly formed during the roasting process, and then the melt wrapped on the surface of SnO_2_ and Na_2_CO_3_, which restrained the formation of Na_2_SnO_3_. It is concluded that Na_8_SnSi_6_O_18_ has a negative impact on the formation of sodium stannate from cassiterite. Next, the formation mechanism of Na_8_SnSi_6_O_18_ is discussed.

### 3.2. Effect of Roasting Atmospheres on the Formation of Na_8_SnSi_6_O_18_

To investigate the formation mechanism of Na_8_SnSi_6_O_18_ in the SnO_2_–SiO_2_–Na_2_CO_3_ system, AR reagents of SnO_2_, SiO_2_ and Na_2_CO_3_ were mixed at a mole ratio of Na_8_SnSi_6_O_18_. The XRD patterns of the samples roasted at 875 °C under a 15 vol.% CO–CO_2_ atmosphere and air atmosphere are shown in Figure 4.

As shown in Figure 4a, the diffraction peaks of Na_8_SnSi_6_O_18_ appeared remarkably, and SnO_2_ and SiO_2_ disappeared at 10 min under a CO–CO_2_ atmosphere. The diffraction peak intensities of Na_8_SnSi_6_O_18_ increased gradually as the roasting time increased from 10 min to 30 min, while those of Na_2_SiO_3_ and Na_2_CO_3_ weakened. However, the phase compositions of the roasted products in an air atmosphere were significantly different, as shown in Figure 4b, and the generation of Na_8_SnSi_6_O_18_ in air was much slower than that in a CO–CO_2_ atmosphere. Moreover, it was noteworthy that no diffraction peaks of Na_2_SnO_3_ were found in the roasted product. Our previous studies illustrated the enhancement of the CO–CO_2_ atmosphere on the formation of Na_2_SnO_3_, as shown in Equation (2). Hence, it can be inferred that Na_2_SnO_3_ may be an important intermediate during the formation of Na_8_SnSi_6_O_18_, as shown in Equation (3). Based on the above phase analysis, in a CO–CO_2_ atmosphere, there were stronger diffraction peaks of Na_8_SnSi_6_O_18_ than in an air atmosphere. The results demonstrated that the formation of Na_8_SnSi_6_O_18_ in the CO–CO_2_ atmosphere was much easier than that in the air atmosphere.
Na_2_CO_3_ + SnO_2_ = Na_2_SnO_3_ + CO_2_(2)
SnO_2_ + 6SiO_2_ + 4Na_2_CO_3_ = Na_8_SnSi_6_O_18_ + 4CO_2_(3)

FTIR analysis was utilized to illustrate the phase transformation of SnO_2_, SiO_2_ and Na_2_CO_3_ roasted products under a 15 vol.% CO–CO_2_ atmosphere, as depicted in Figure 5. 

The peak at 1485 cm^−1^ was assigned to Si=O stretching vibrations, and vibrations at 1076 cm^−1^ and 932 cm^−1^ correspond to Si-O-Si bonds [25,26]. The results in Figure 5a,b indicate that the intensity of the Si=O bond decreased obviously as the roasting temperature and roasting time increased, which revealed the conversion of SiO_2_. Simultaneously, the increase in Si-O-Si bonds and Sn-O-Si/Si-O-T bonds (614 cm^−1^, 545 cm^−1^ and 449 cm^−1^) [25,26,27,28] can also be observed in Figure 5. The results further confirmed the molecular evolution of Si-bearing materials during the roasting process, which was consistent with the XRD analysis in Figure 4.

### 3.3. Effect of Intermediate Products on the Formation of Na_8_SnSi_6_O_18_

The results in Figure 4 demonstrate that Na_2_SnO_3_ and Na_2_SiO_3_ were generated along with Na_8_SnSi_6_O_18_, as shown in Equations (2) and (4), then both intermediate products were taken into consideration to reveal the reaction path for the formation of Na_8_SnSi_6_O_18_. SnO_2_ or SiO_2_ was first mixed with Na_2_CO_3_ at a mole ratio of 1:1 and then roasted at 875 °C under a 15 vol.% CO–CO_2_ atmosphere for a certain period of time. The XRD analysis of the roasted products is shown in Figure 6.
Na_2_CO_3_ + SiO_2_ = Na_2_SiO_3_ + CO_2_(4)

As shown in Figure 6a, the reaction between SnO_2_ and Na_2_CO_3_ (Equation (2)) proceeded much more quickly; Na_2_SnO_3_ was the main phase in the roasted products, and almost no diffraction peaks of SnO_2_ were found after roasting for 15 min. In contrast, the formation rate of Na_2_SiO_3_ was slow in the solid-state, and the diffraction peaks were uncertain in Figure 6b after roasting for 30 min. The difference in the solid-state reaction rates of Equations (2) and (4) may cause different reaction paths for the final products; therefore, two possible reactions were proposed, as shown in Equations (5) and (6) based on conservation of mass.
Na_2_SnO_3_ + 6SiO_2_ + 3Na_2_CO_3_ = Na_8_SnSi_6_O_18_ + 3CO_2_(5)
6Na_2_SiO_3_ + SnO_2_ = Na_8_SnSi_6_O_18_ + 2Na_2_O(6)

In view of further verification, two kinds of mixed samples were prepared as follows: Na_2_SnO_3_·3H_2_O, Na_2_CO_3_ and SiO_2_ (AR reagent) with a molar ratio of 1:3:6, as shown in Equation (5), and Na_2_SiO_3_·3H_2_O and SnO_2_ with a molar ratio of 6:1 as shown in Equation (6). Then, TG-DSC (Ar atmosphere, ~1000 °C) and XRD analysis were used to determine the possible reactions in the two designed systems of Na_2_SnO_3_–SiO_2_ and Na_2_SiO_3_–SnO_2_, and the results are displayed in Figure 7 and Figure 8, respectively.

As shown in Figure 7a, two weak endothermic peaks at 87.5 °C and 243.8 °C were observed with a small quantity of weight loss, which was assigned to the thermal dehydration reaction of Na_2_SnO_3_·3H_2_O and Na_2_CO_3_ (crystal water) [29]. After that, the mass loss increased sharply to 18.3 wt.% with a significant endothermic peak at 833.6 °C in the DSC curve. The results revealed that the solid-phase reaction proceeded in the temperature range of 800–900 °C, the weight loss was possibility attributed to the reaction of Equation (5) and released CO_2_ gas. In addition, the XRD results in Figure 7b indicated that almost no diffraction peak of Na_2_SnO_3_ can be found, which illustrates that Na_2_SnO_3_ in the raw materials is converted to Na_8_SnSi_6_O_18_ completely, as shown in Equation (5). On the other hand, the results in Figure 8 show totally different outcomes. No exothermic reactions were found in the TG-DSC curve (in Figure 8a), and the roasted products were unchanged as SnO_2_ and Na_2_SiO_3_ (in Figure 8b), which excluded the reaction paths expressed in Equation (6).

### 3.4. Reactions between Na_8_SnSi_6_O_18_ and Na_2_CO_3_

To find a possible transition process of Na_8_SnSi_6_O_18_ during the roasting process, Na_8_SnSi_6_O_18_ was synthesized based on our previous study [21]. In this section, Na_2_CO_3_: SnO_2_: SiO_2_ were mixed as mole ratio of 4:1:6, with a roasting temperature of 1000 °C and roasting time of 360 min. The XRD pattern of synthetic Na_8_SnSi_6_O_18_ is shown in Figure 9a. The synthesized Na_8_SnSi_6_O_18_ was well matched with the PDF standard card of No. 85-0532, and there were no diffraction peaks of impurities. The TG-DSC analysis of Na_8_SnSi_6_O_18_ is given in Figure 9b. Na_8_SnSi_6_O_18_ was stable during the heating process, while a phase transition occurred in the temperature range of 800–850 °C with an endothermic peak at 825.6 °C in the DSC curve. A further test to determine the melting behavior of Na_8_SnSi_6_O_18_ using in situ high temperature thermal analysis is shown in Figure 9b. The results showed that the structure started to change when the temperature reached 800 °C, and a small amount of liquid was formed at this moment. The sample was almost fully molten into the liquid phase as the temperature increased to 825 °C. The results verified the endothermic peak in the DCS curve and corresponded to the melting point of Na_8_SnSi_6_O_18_.

Based on the above results, it was found that the mole ratio of Na in Na_8_SnSi_6_O_18_ was much lower than that of Na_2_SiO_3_/Na_2_SnO_3_, as shown in Equations (2) and (4). The reactions between Na_8_SnSi_6_O_18_ and Na_2_CO_3_ were discussed in the case of excess Na_2_CO_3_ dosage. Then, Na_8_SnSi_6_O_18_ and Na_2_CO_3_ were mixed at a mole ratio of 1:5, and TG-DSC analysis was conducted. The TG-DSC results and the XRD patterns of the roasted products are shown in Figure 10.

According to Figure 10a, obvious weight loss started from 800 °C to 950 °C, and two endothermic peaks in the DSC curve were found at 823 °C and 851 °C. The melting point of Na_2_CO_3_ was 851 °C, and it can be inferred that the mass loss was attributed to Na_2_CO_3_ decomposition in the presence of Na_8_SnSi_6_O_18_. It is noteworthy that, as found in Figure 10b, that Na_2_SiO_3_ and Na_2_SnO_3_ were the main phases in the final TG products, and no diffraction peak for Na_8_SnSi_6_O_18_ remained, which indicated that Na_8_SnSi_6_O_18_ easily reacted with excess Na_2_SnO_3_. A possible reaction was proposed, as shown in Equation (7), based on conservation of mass.
Na_8_SnSi_6_O_18_ + 3Na_2_CO_3_ = 6Na_2_SiO_3_ + Na_2_SnO_3_ + 3CO_2_(7)

To verify the above analysis, the effect of roasting temperature and roasting time on the phase transformation of Na_8_SnSi_6_O_18_ was investigated, and the Na_2_CO_3_/Na_8_SnSi_6_O_18_ mole ratio was fixed at 3:1, as in Equation (6). Figure 11 shows the XRD patterns of the Na_2_CO_3_/Na_8_SnSi_6_O_18_ mixed samples roasted in the temperature range of 800–900 °C with a time of 30–120 min.

As shown in Figure 11a, the main phase in the roasted products was unchanged as Na_8_SnSi_6_O_18_ at 800 °C, while the diffraction peaks of Na_2_SiO_3_ and Na_2_SnO_3_ were uncertain. The structural diffraction peaks of Na_8_SnSi_6_O_18_ weakened and then vanished when the temperature exceeded 850 °C. Figure 11b shows that the diffraction peaks of Na_2_SiO_3_ appeared and Na_8_SnSi_6_O_18_ decreased at 30 min, and then the peaks of Na_8_SnSi_6_O_18_ gradually decreased and disappeared as the roasting time was prolonged to 90 min and 120 min. Na_2_SiO_3_ and Na_2_SnO_3_ were the final roasted products expressed as Equation (7).

### 3.5. Discussion on the Reaction Mechanism of the SnO_2_–SiO_2_–Na_2_CO_3_ System

During the soda-roasting process of cassiterite concentrates, the overriding aim was to synchronously promote the transformation of stubborn minerals (SnO_2_ and SiO_2_) into freely soluble materials (Na_2_SnO_3_ and Na_2_SiO_3_). However, Na_8_SnSi_6_O_18_ was inevitably generated during the roasting process, which was almost insoluble in the leaching process and markedly decreased the recovery of Sn. Based on the above results and our previous studies, the reaction mechanism of the SnO_2_–SiO_2_–Na_2_CO_3_ system and the formation of Na_8_SnSi_6_O_18_ can be summarized as follows in Figure 12.

First, SnO_2_ reacted with Na_2_CO_3_ to form Na_2_SnO_3_ as shown in Equation (2). Meanwhile, part of SiO_2_ also reacted with Na_2_CO_3_ to form a small amount of Na_2_SiO_3_, see Equation (4). Nonetheless, the reaction rate was much lower than that of Equation (2). Then, Na_2_SnO_3_ reacted immediately with Na_2_CO_3_ and SiO_2_ to form Na_8_SnSi_6_O_18_ as shown in Equation (5), and Na_2_SnO_3_ was the key intermediate during the formation of Na_8_SnSi_6_O_18_, while the reaction between Na_2_SiO_3_ and SnO_2_ was impossible. The melting point of Na_8_SnSi_6_O_18_ was measured at 825 °C, which was much lower than that of other materials in the SnO_2_–SiO_2_–Na_2_CO_3_ system. Thus, Na_8_SnSi_6_O_18_ was always formed accompanied by SnO_2_ and invariably closely wrapped around SnO_2_ particles, which blocked the contact between SnO_2_ and Na_2_CO_3_. Therefore, the formation of Na_2_SnO_3_ was significantly inhibited once Na_8_SnSi_6_O_18_ was present. In addition, Na_8_SnSi_6_O_18_ is an unstable compound that can react with excess Na_2_CO_3_ (Equation (7)) as the roasting temperature and time increase.

## 4. Conclusions

During the process of sodium stannate preparation from cassiterite concentrate under a CO–CO_2_ atmosphere, the formation of Na_8_SnSi_6_O_18_ in the SnO_2_–SiO_2_–Na_2_CO_3_ system affects the quality of sodium stannate products. To solve this problem, the effects of Na_8_SnSi_6_O_18_ formation on the product quality were investigated in this study, and the following conclusions were obtained:The reactions between Na_2_CO_3_ and SnO_2_/SiO_2_ proceeded simultaneously during the roasting process, while the formation of Na_2_SnO_3_ was promoted under a CO–CO_2_ atmosphere. Then, Na_8_SnSi_6_O_18_ was easily formed once Na_2_SnO_3_ appeared; nonetheless, the reaction between Na_2_SiO_3_ and SnO_2_ was impossible;The melting point of Na_8_SnSi_6_O_18_ is only 825 °C, which is much lower than that of Na_2_CO_3_, Na_2_SnO_3_ and Na_2_SiO_3_ in the SnO_2_–SiO_2_–Na_2_CO_3_ system. Na_8_SnSi_6_O_18_ closely wrapped around the SnO_2_ particles and restrained the reaction between SnO_2_ and Na_2_CO_3_;Na_8_SnSi_6_O_18_ is an unstable compound, and the reaction between Na_8_SnSi_6_O_18_–Na_2_CO_3_ can proceed as Na_8_SnSi_6_O_18_ + 3Na_2_CO_3_ = 6Na_2_SiO_3_ + Na_2_SnO_3_ + 3CO_2_. The reaction was controlled by higher temperatures of above 800 °C as the roasting time was prolonged.

## Figures and Tables

**Figure 1 materials-15-08718-f001:**
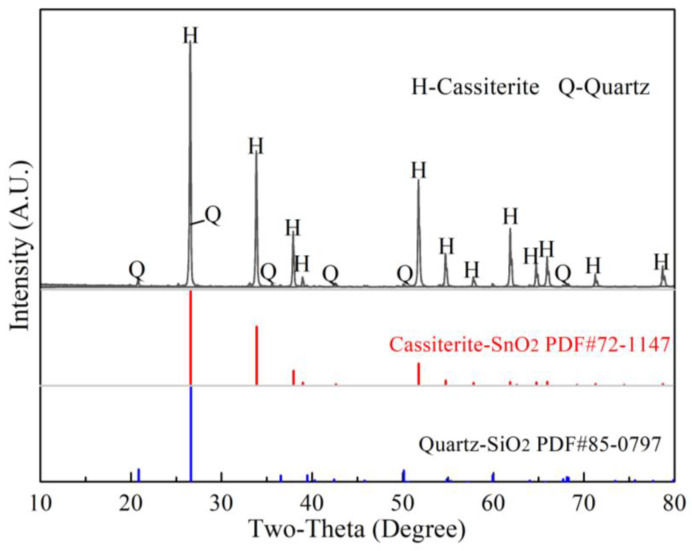
XRD patterns of Si-bearing cassiterite concentrates.

**Figure 2 materials-15-08718-f002:**
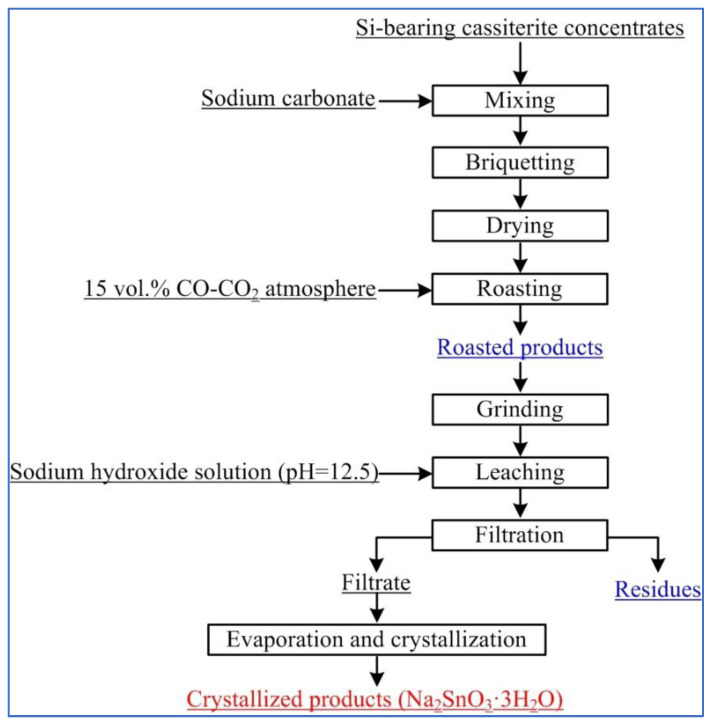
The flowsheet for preparing sodium stannate from Si-bearing cassiterite concentrates.

**Figure 3 materials-15-08718-f003:**
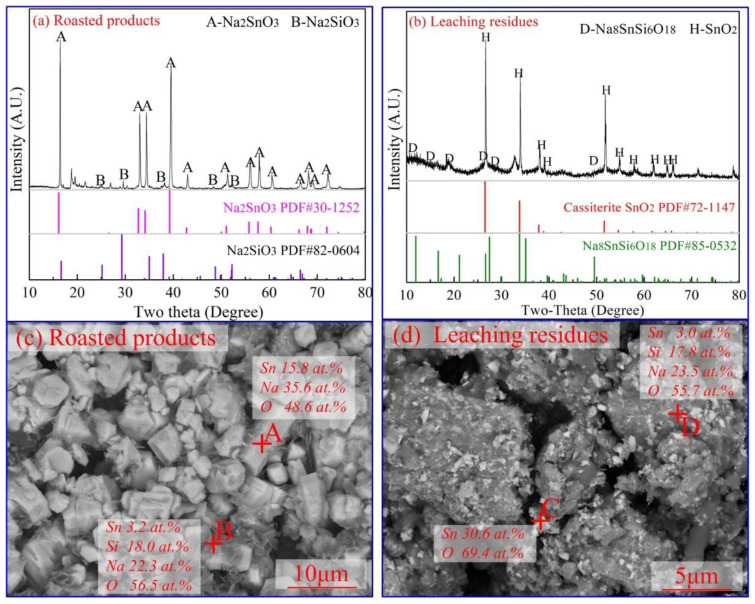
Phase analysis of the roasting products and the leaching residues. ((**a**)-XRD results of roasted products, (**b**)-XRD results of leaching residues, (**c**)-SEM analysis of the roasted products, (**d**)-SEM analysis of leaching residues).

**Figure 4 materials-15-08718-f004:**
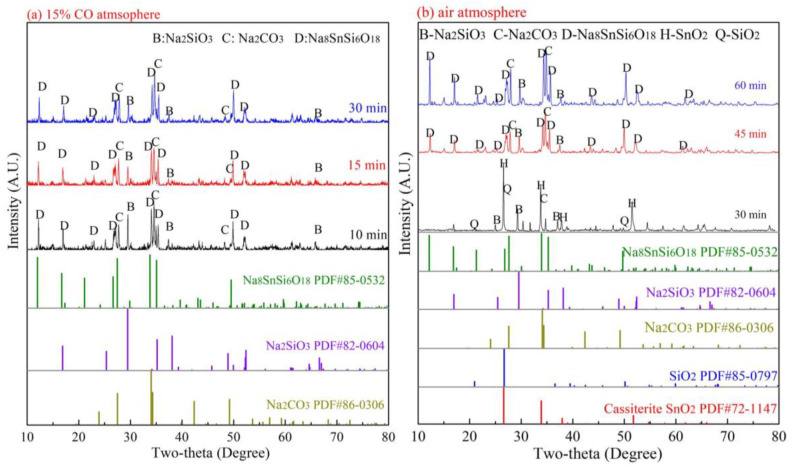
Effect of atmosphere on the formation of Na_8_SnSi_6_O_18_ (at 875 °C).

**Figure 5 materials-15-08718-f005:**
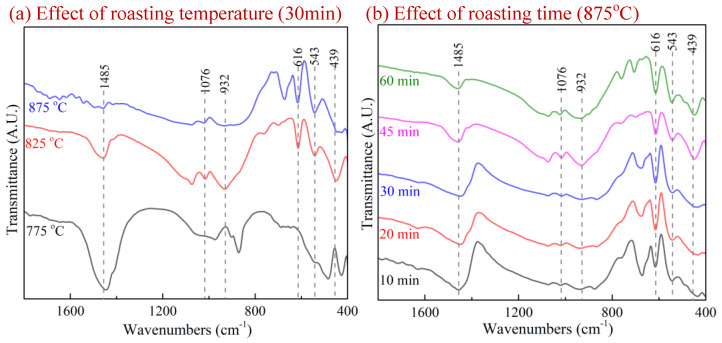
FTIR spectra of SnO_2_, SiO_2_ and Na_2_CO_3_ roasted products under 15 vol.% CO–CO_2_ atmosphere.

**Figure 6 materials-15-08718-f006:**
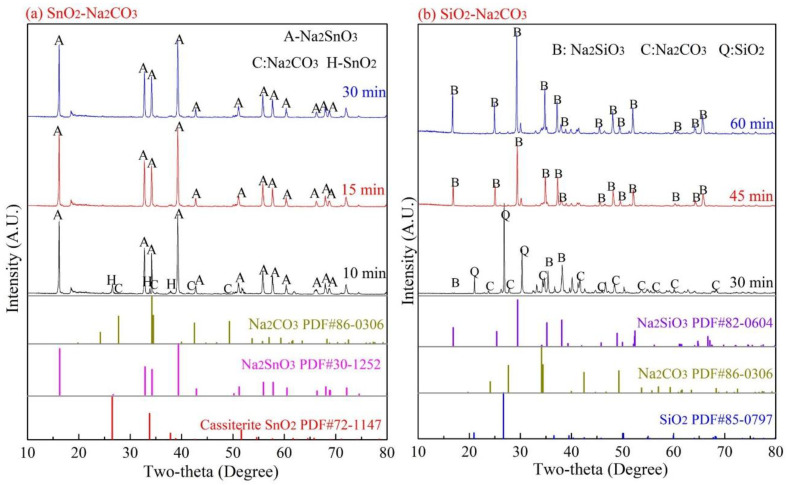
Reactions of SnO_2_–Na_2_CO_3_ and SiO_2_–Na_2_CO_3_ systems (875 °C).

**Figure 7 materials-15-08718-f007:**
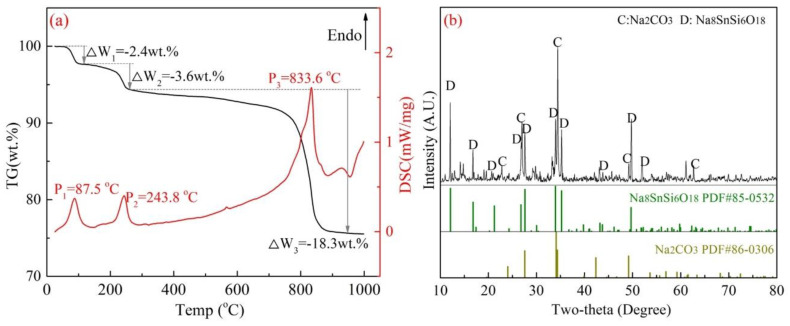
Reactions between Na_2_SnO_3_·3H_2_O and SiO_2_ ((**a**)-TG-DSC analysis, (**b**)-XRD analysis of the TG products).

**Figure 8 materials-15-08718-f008:**
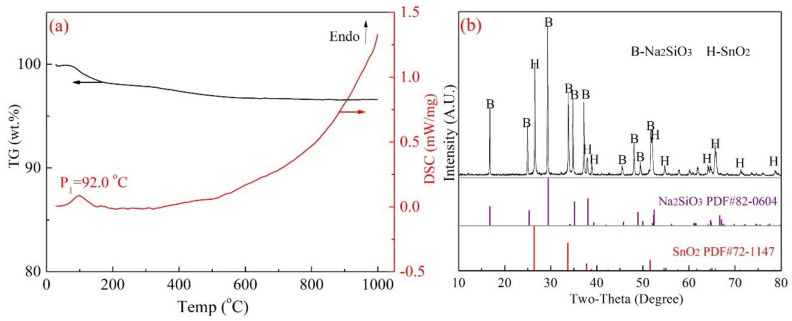
Reactions between Na_2_SiO_3_ and SnO_2_ ((**a**)-TG DSC analysis, (**b**)-XRD analysis of the TG products).

**Figure 9 materials-15-08718-f009:**
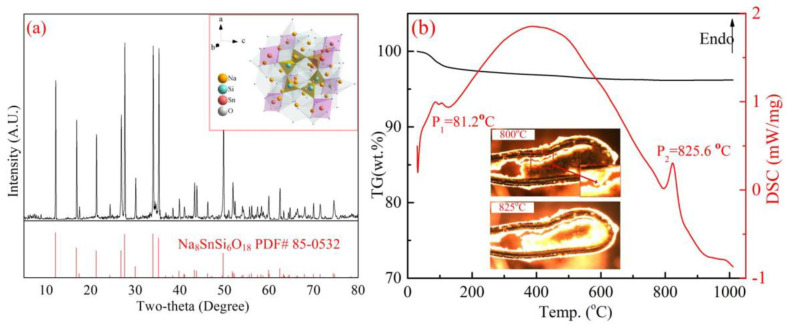
Properties of synthetic Na_8_SnSi_6_O_18_ ((**a**)-XRD patterns, (**b**)-TG-DSC analysis).

**Figure 10 materials-15-08718-f010:**
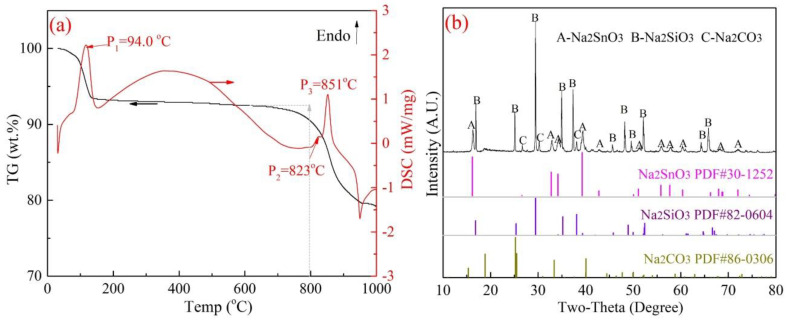
Reactions between Na_8_SnSi_6_O_18_ and Na_2_CO_3_ ((**a**)-TG-DSC analysis, (**b**)-XRD analysis of the TG products).

**Figure 11 materials-15-08718-f011:**
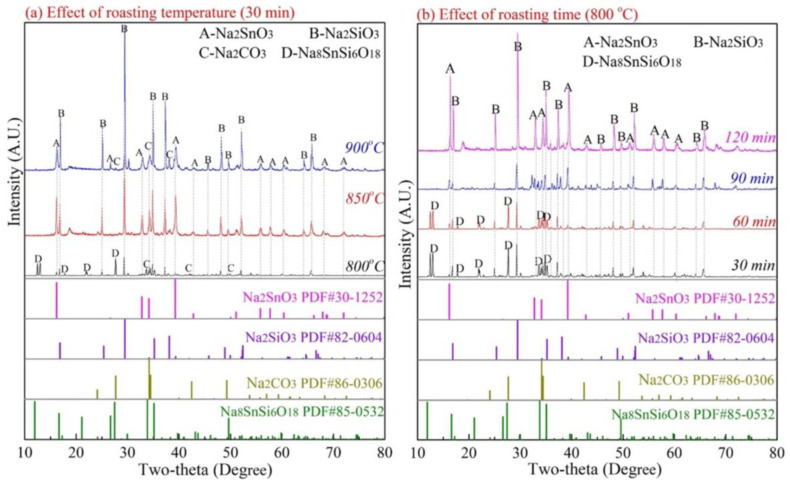
Effect of roasting temperature and time on the reaction between Na_8_SnSi_6_O_18_ and Na_2_CO_3_.

**Figure 12 materials-15-08718-f012:**
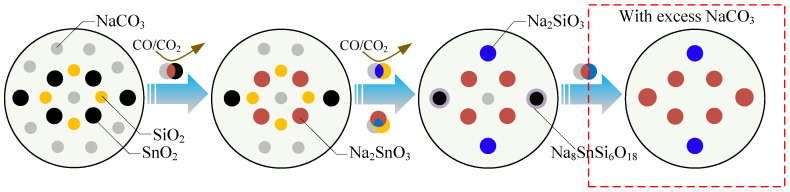
Phase evolution in the SnO_2_–SiO_2_–Na_2_CO_3_ system during soda roasting process.

## Data Availability

The data presented in this study are available on request from the corresponding author.

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
