# Peer review of "Formation and Inhibition Mechanism of Na8SnSi6O18 during the Soda Roasting Process for Preparing Na2SnO3"

_materials, 2022, doi:10.3390/ma15248718_

Round 1

Reviewer 1 Report

Review of the manuscript (materials-1987644) entitled

" Improving the quality of Na2SnO3 prepared from cassiterite concentrates by regulating the transformation of Na8SnSi6O18"

By Su et al.

In this manuscript, the authors developed novel process for the preparation of sodium stannate, that widely used in ceramics and electroplating industries, from cassiterite concentrates. The result of their great work is remarkable. The manuscript is well written, organized, fluent, and to the point. I think that the manuscript looks more suitable for an international audience.

My only comments are:

1-    I suggest for the authors to write short paragraph about and their host natural rocks.

2-    The target of the study, at the end of the introduction, is poorly written.

3-    Fixing the grammars and language typesetting errors (see the attached pdf file)

I believe that the manuscript would have the quality of being accepted after minor revision for publication in Materials.

Reviewer 2 Report

...

The manuscript entitled "Improving the quality of Na2SnO3 prepared from cassiterite concentrates by regulating the transformation of Na8SnSi6O18" by Zijian Su, Shuo Liu, Benlai Han, Yuanbo Zhang, Tao Jiang is devoted to the study of the reasons for the low yield of sodium stannate due to the formation of Na8SnSi6O18 for the purpose of mineral processing - cassiterite. The content of the manuscript corresponds to the subject of the special issue of the journal "Materials", but it is also possible to publish it in journals of other subjects, such as "Minerals". The reviewer recommends that the authors make some corrections and improvements so that the work can be accepted for publication.

Below I will give some comments that arose during acquaintance with the manuscript. Please do not engage in correspondence and polemics with the reviewer - all answers must be, one way or another, reflected in the text of the manuscript.

General remarks.

The paper lacks data from thermodynamic calculations. Most of the time-consuming experiments could have been avoided by a preliminary analysis of the thermodynamic properties of the components and their phase diagrams. Now this area of science is very well developed and accessible.  It is simply strange today to determine the direction of reactions between Na2SiO3 and SnO2 by direct experiment. This is a student problem in thermodynamics. Why did the authors ignore the standard approach: first thermodynamic analysis, then experimental confirmation?

The choice of the synthesis atmosphere is not clear. On the one hand, since carbon dioxide is one of the decomposition products of sodium carbonate, an increase in its partial pressure should shift the equilibrium towards the original products (see comments below). On the other hand, the presence of carbon monoxide is completely unjustified. He is poisonous. It is a reducing agent, and theoretically it can reduce tin. What is the point in such a gaseous atmosphere? Why is this choice better than, for example, nitrogen?

The authors chose X-ray diffractometry as the main method for studying the synthesis products. I note that, since the main objects of study in this work are tin-containing compounds, in order to obtain complete information about their transformation, it would be possible to use the method of Mössbauer spectroscopy.

The authors used apparently not pure starting materials. This is clearly seen from the mass loss data. The work would have looked better if the authors had previously dried the reagents before the study. This is not difficult, but it greatly changes the reader's attitude to the quality of the work.

All diffractograms: Modify the drawings so that the reader can see the experimental data. They overlap with letters and lines. Use the full width of the page. Each X-ray diffraction pattern (not just Figure 9) should have the numbers of the standard ICDD PDF charts that you used in your interpretation.

All figures - Figure captions end with a dot.

56, 61 "cassiterite concentrates", "AR reagents of SnO2, SiO2 and Na2CO3" Give details of the origin of the starting products.

65 Provide links to the specific data of the ICDD PDF-2 database that you used when interpreting the diffraction pattern. Were all reflexes interpreted properly?

74 Equation on a separate line. Replace the dot with a comma.

80 Was an X-ray filter used?

84 Specify the technique for obtaining spectra: transmission or reflection?

86 Specify the material of the crucibles and the weight of the samples.

89 Indicate how the sample preparation was carried out.

104 Figure captions must end with a dot. Add information about parts a, b, c, e to the figure caption.

130-132 "The diffraction peak intensities of Na8SnSi6O18 increased gradually as the roasting time increased from 10 min to 30 min, while those of Na2SiO3 and Na2CO3 weakened" The figure does not show any noticeable changes. In addition to reducing B from 10 to 15 minutes. Could the authors offer a quantitative description of the change in the diffractograms to confirm their statements?

138-139 "that Na2SnO3 may be an important intermediate during the formation of Na2SnO3, as shown in Eq. (2)." In (2) there is no reagent Na2SnO3. How can Na2SnO3 be an intermediate of itself?

140 "stronger and more diffraction peaks of" Do the authors mean the number of peaks, their intensity, width or relative area? Have the authors tried to carry out a quantitative analysis of the composition of the material based on the X-ray diffraction data?

(1) Since carbon monoxide does not participate in the indicated reaction equations, what role do the authors assign to it when carrying out the reaction in an atmosphere of such a complex and dangerous composition for the health of experimenters?

148 "corresponds to Si-O-Si bonds." Please provide a link to relevant sources of information on the interpretation of absorption bands.

162 "(Degree)"

163 In figure (a) 10 min there are reflections of unreacted SnO2, but no sodium carbonate. Why?

What is indicated by the letter "B" in figure (b)?

166 "10 min" - - > "15 min"

175 "Na2SnO3-SiO2 and Na2SiO3" Here, experiments are carried out with reagents not specified in the methodological part of the manuscript. What is their origin and what is known about their purity?

178 Endothermic effects are usually pointed down. (a) Does your hypothesis agree with the change in mass at temperatures of 88, 244, and 834°C? (b) The diffraction pattern is depicted in such a way that it is not clear where the reflections themselves are, and where the lines from the pointers are. Please provide a good image so that the reflexes can be clearly seen.

181 Please provide a link to literary sources that would confirm your hypothesis about the removal of water of crystallization at 244°C.

183 "Na2SnO3 (crystal water)" Why not sodium carbonate (hydrocarbonate)? See Figure 10 for comparison.

186 "the only possibility of weight loss was attributed" Support your hypothesis with literature data.

186 "thermal decomposition of Na2CO3" So sodium carbonate was present in the system? Please indicate this explicitly in the text and captions to the figures above.

192 " which excluded the reaction paths expressed as Eq. (5). " Why did the authors not show this by thermodynamic calculations?

195 "Na8SnSi6O18 was synthesized based on our previous study [17]," In [17], there is no procedure for the synthesis of this preparation. Please provide a detailed synthesis procedure here so that it can be reproduced.

206 "the melting point Na8SnSi6O18." To confirm the phase transition, it is necessary to obtain a cooling curve, where an exoeffect would be observed with a true phase transition.

208 "Thermal property " --> "Property "

232 Normalize diffractogram images so that they have the same intensity. Arrange the peak labels so as not to overlap the experimental data.

250-251 " which was encouraged by a CO-CO2 atmosphere," This is a false statement, because according to Le Chatelier's principle, an increase in the partial pressure of the reaction product shifts the equilibrium towards the starting materials. https://en.wikipedia.org/wiki/Le_Chatelier%27s_principle

258-259 " which blocked the contact between SnO2 and Na2CO3" Quite strange reasoning... 1. If there was no contact between the particles, then the addition of a liquid component cannot interfere with it. If it was, then the same. 2. At such high temperatures, the ion diffusion coefficients are large enough to overcome the "liquid barriers". 3. The very fact that the reactants come into contact with the melt should contribute to the acceleration of reactions, since diffusion through the melt is much simpler. Compare reaction rates in solutions and between crystalline substances. Why do you think the former are much larger?

264 "of the final product" The authors did not give practical procedures for the synthesis of sodium stannate. Until now, the formation of an anhydrous compound has been discussed, and in the final, the authors present data for the trihydrate, the synthesis of which has not been described. Other properties of the final product, apart from the diffraction pattern, are not described. Those. the goals stated in the title of the manuscript about "Improving the quality of Na2SnO3" have not been achieved, since it is not possible for the reader to characterize and compare the quality of the product.

266-267 "standard of the industrial first-grade stannate (GB/T26040-2010)." What characteristics did the authors define for the final product in order to claim that it complies with the standard of the industrial firstrade stannate (GB/T26040-2010)? Why do the authors not provide this data here?

...

Round 2

Reviewer 2 Report

...

The main remaining problem with the manuscript is the crumpled ending. Authors should either remove references to "product" or give a broad description of what it is and where it came from in this work.

Remarks (Authors should take note and make corrections)

" Response 25:"

R2. You probably forgot to include this reference in the manuscript?

" Point 36: 264 "of the final product" The authors did not give practical procedures for the synthesis of sodium stannate. ...

Response 36: Based on a previous paper, ..."

R2: Let's repeat it again. In 289 you write about sodium stannate obtained by calcination at 800°C. And in 291 you are already discussing a certain “product” that you don’t know how you got, and this “product” seems to be sodium stannate crystalline hydrate. Where is the transition from one substance to another described?

65-69 Did you add the conclusions of the manuscript to the introduction?

72 Could you give a more precise indication, the name of the deposit, geolocation coordinates?

78 "Na2SiO3 and Na2SnO3•3H2O" Is there an error here? Or then an error in 197 and/or 198?

295 " Based on a previous paper" []?

299" The Sn leaching efficiency was increased to 99.2%." As a result of what manipulations described in this manuscript did these data appear? Or is the methodology for determining Sn leaching efficiency described somewhere in this manuscript?

301 What %?

322 " from 85.6% to" These data were not obtained in this work, were they? Why do you indicate them in the conclusions of this manuscript?

Discussion (Authors can ignore and not reply)

"Point 1: The paper lacks data from thermodynamic calculations....

Response 1: ...We found that CO-CO2 atmosphere has a significant effect on the reactions of Sn-bearing materials. Then the pahse transfomation mechanisms of stannates can not be analysed based on troditional thermodynamic analysis method. "

R2: Why? Traditional methods of thermodynamic analysis work in all atmospheres.

"In addition, synthesis reactions were mainly proceeded in solid-phase, and no phase diagrams and G-T data (Na8SnSi6O18) can be found. Hence, in this study, we used the basic methods to investigate the transition behaviors in SnO2-SiO2-Na2CO3 system. "

R2: I admit that for Na8SnSi6O18 there are difficulties in determining the thermodynamic data. But for the remaining components and reactions of type (2), (4), the data are sufficient for theoretical analysis. Right? In addition, I am sure that many variants of the phase diagrams of the SnO2-SiO2-Na2O and SnO2-SiO2-Na2CO3 systems are known.

"Point 2: The choice of the synthesis atmosphere is not clear..."

Response 2: ... the reaction mechanism can be expressed as follows: ..."

R2: This discussion is not relevant to this manuscript. You link to your other works. I did not watch them, I read your conclusions. I have to disappoint you - your approaches are not correct. a) Removal of "bridging oxygens" is a tin reduction process. b) The activity of oxygen anions in sodium oxide is no different from the activity of oxygen anions in tin oxide. c) You are trying to hide the shortcomings of your hypothesis with scientific equations, but your focus is easily revealed if you are asked to carefully write ALL reaction equations in compliance with the material and electrical balance. For some reason, in the figure you have the oxygen ionization equation - where do the electrons come from? Where is the oxygen from? Next, carbon dioxide and an electron are formed. Where does the electron go? But this discussion has nothing to do with this manuscript.

" Point 4: The authors used apparently not pure starting materials

Response 4: The raw materials used in this paper "

R2. My remark was not about cassiterite, but about "analytical reagents". Your sodium carbonate contains water or sodium bicarbonate impurities. Or both.

" Point 24: 178 ...(a) Does your hypothesis agree with the change in mass at temperatures of 88, 244, and 834 °C? (

Response 24: ... The Na2SnO3 sample used in this sdudy were AR Na2SnO3·3H2O,..."

R2. Why then do you write Na2SnO3 in your manuscript? Anhydrous salt and crystalline hydrate are completely different substances. Even I am already confused and do not understand where we are talking about hydrate, and where about anhydrous salt.

"... The mass loss aroound 244 oC can be attributed to the transformation of Na2SnO3·3H2O to Na2SnO3. In addition, the mass loss around 834 oC belonged to the emission of CO2. "

R2. I'm asking: does the weight loss of 3.6% at 244°C correspond to 3H2O? And 18.3% loss of CO2 based on the ratio of components?

" Point 35: 258-259 " ... Quite strange reasoning...

Response 35: Based on our previous studies ... Hence, we make the following inference in 258-259. "

R2: I am writing about the generally accepted concepts of diffusion in melts at high temperatures, and you are answering about the magical properties of a gas mixture, the mechanism of action of which seems to be unknown. This is not a discussion; this is an avoidance of an answer.

...

Author Response

Thanks again for the reviewers’ positive comments and constructive suggestions on our manuscript entitled “Improving the quality of Na2SnO3 prepared from cassiterite concentrates by regulating the transformation of Na8SnSi6O18”. We have studied your advice and the reviewers’ comments carefully and made revisions marked in red in the revised manuscript. The details of our revisions and responses are as following.

The main remaining problem with the manuscript is the crumpled ending. Authors should either remove references to "product" or give a broad description of what it is and where it came from in this work.

Response: Thanks for your suggestion. This study focused on the transformation of Na8SnSi6O18 during the soda roasting process. Hence, we removed the references to "product" in the end of the manuscript, to prevent misleading the readers.

Remarks (Authors should take note and make corrections)

"Response 25:" R2. You probably forgot to include this reference in the manuscript?

Response: We have added it in the revised manuscript. [29] T.G. Santos, A.O.S. Silva, S. M.P. Meneghetti, Comparison of the hydrothermal syntheses of Sn-magadiite using Na2SnO3 and SnCl4·5H2O as the precursors, Applied Clay Science. 183 (2019) 105293, https://doi.org/10.1016/j.clay.2019.105293.

"Point 36: 264 "of the final product" The authors did not give practical procedures for the synthesis of sodium stannate. ...Response 36: Based on a previous paper, ..." R2: Let's repeat it again. In 289 you write about sodium stannate obtained by calcination at 800°C. And in 291 you are already discussing a certain “product” that you don’t know how you got, and this “product” seems to be sodium stannate crystalline hydrate. Where is the transition from one substance to another described?

Response: Based on our series researches, we found that the formation of Na8SnSi6O18 during soda roasting process was unavoidable, which decreased the leaching efficiency of Sn. However, the formation mechanism of Na8SnSi6O18 was seldom reported. Then, in this work, we payed more attention to the formation reaction path of Na8SnSi6O18. As your suggestion, we have removed the discussing on the leached-crystallized product. And the title was revised as “Formation and inhibition mechanism of Na8SnSi6O18 during the soda roasting process of preparing Na2SnO3”.

65-69 Did you add the conclusions of the manuscript to the introduction?

Response: We have modified the introduction.

72 Could you give a more precise indication, the name of the deposit, geolocation coordinates?

Response: The cassiterite concentrates (taken from Gejiu, Yunnan Province of China, Yunnan Tin Company Limited) used in this study.

78 "Na2SiO3 and Na2SnO3·3H2O" Is there an error here? Or then an error in 197 and/or 198?

Response: Only analytical reagents of Na2SnO3·3H2O can be obtained. We have revised the mistakes in the revised manuscript.

295 "Based on a previous paper" []? 299 "The Sn leaching efficiency was increased to 99.2%." As a result of what manipulations described in this manuscript did these data appear? Or is the methodology for determining Sn leaching efficiency described somewhere in this manuscript? 301 What %? 322 "from 85.6% to" These data were not obtained in this work, were they? Why do you indicate them in the conclusions of this manuscript?

Response: Thanks for your carefully revision, we removed the references to "product" in the end of the manuscript as your suggestion, to help the readers better understand the core idea of this study.

Special thanks to the reviewers’ good suggestions. Now, I believe the quality of the present state should meet the requirement of this journal. I also think the readers should understand what we have done. We appreciate for your warm work and thank you again for your help! We sincerely hope that the corrections will meet with approval and looking forward to hearing from you soon!

Best regards for you!

Sincerely yours

Yuanbo Zhang
